# Design of Computational Models for Hydroturbine Units Based on a Nonparametric Regression Approach with Adaptation by Evolutionary Algorithms

**Vladimir Viktorovich Bukhtoyarov [1,2] and Vadim Sergeevich Tynchenko [1,2,***

1   Department of Technological Machines and Equipment of Oil and Gas Complex, School of Petroleum and Natural Gas Engineering, Siberian Federal University, 660041 Krasnoyarsk, Russia; vladber@list.ru
2   Information-Control Systems Department, Institute of Computer Science and Telecommunications, Reshetnev Siberian State University of Science and Technology, 660037 Krasnoyarsk, Russia
\*   Correspondence: vadimond@mail.ru; Tel.: +7-95-0973-0264

**Abstract:** This article deals with the problem of designing regression models for evaluating the parameters of the operation of complex technological equipment—hydroturbine units. A promising approach to the construction of regression models based on nonparametric Nadaraya–Watson kernel estimates is considered. A known problem in applying this approach is to determine the effective values of kernel-smoothing coefficients. Kernel-smoothing factors significantly impact the accuracy of the regression model, especially under conditions of variability of noise and parameters of samples in the input space of models. This fully corresponds to the characteristics of the problem of estimating the parameters of hydraulic turbines. We propose to use the evolutionary genetic algorithm with an addition in the form of a local-search stage to adjust the smoothing coefficients. This ensures the local convergence of the tuning procedure, which is important given the high sensitivity of the quality criterion of the nonparametric model. On a set of test problems, the results were obtained showing a reduction in the modeling error by 20% and 28% for the methods of adjusting the coefficients by the standard and hybrid genetic algorithms, respectively, in comparison with the case of an arbitrary choice of the values of such coefficients. For the task of estimating the parameters of the operation of a hydroturbine unit, a number of promising approaches to constructing regression models based on artificial neural networks, multidimensional adaptive splines, and an evolutionary method of genetic programming were included in the research. The proposed nonparametric approach with a hybrid smoothing coefficient tuning scheme was found to be most effective with a reduction in modeling error of about 5% compared with the best of the alternative approaches considered in the study, which, according to the results of numerical experiments, was the method of multivariate adaptive regression splines.

**Keywords:** turbine unit; modeling; nonparametric regression; smoothing coefficient; numerical experiment; adaptation; evolutionary algorithm; optimization

## 1. Introduction

Hydroturbine units (HTUs) are the main generating equipment for a significant number of power plants in Russia and around the world. The technical condition of such units is one of the key parameters for their efficient and safe operation. Taking into account the high criticality of the failures of such equipment, the subject of research is the formation of reliable procedures and systems for evaluating technical condition parameters. Such systems are based on the measurement, assessment, and prediction of a number of characteristics that make it possible to make a decision on the technical condition of the HTU and the parameters of its operating mode.

The highest requirements for operational safety imply the need to build models that provide the most accurate calculation of a set of current parameters. This is because a

number of parameters require evaluation in conditions of insufficient information, and to assess the adequacy of the operation of measuring instruments, taking into account their possible failure or large measurement errors. The output parameters of such models make it possible to comprehensively assess the state of the investigated object (HTU); however, in real operation mode, monitoring some of them is impossible, and some measurements cannot be performed in synchronous mode. This also requires computational models that allow for the synchronization of measured and calculated values, which provides the required sampling level for a complex measuring and control information system. Thus, it seems relevant to develop and evaluate approaches to modeling that provide the highest possible accuracy in estimating parameters for HTUs.

In scientific works in this area, to build models for assessing HTU parameters, it was proposed to use the support-vector method, methods for modeling dynamic modes, a lateral vibration-response calculation model for a rotor-bearing shaft system, control models based on fuzzy logic, the kernel clustering method, probabilistic neural networks, and others [1–13]. In a number of works, from those listed above, the authors managed to form models that provide fairly good accuracy in estimating the parameters of the operation of HTU, including those based on vibration characteristics. Nevertheless, the ever-increasing requirements for the accuracy of computational models in view of their integration into complex production systems form the need to seek alternative and potentially more accurate modeling methods for this problem.

In addition to the requirements for high accuracy in assessing the parameters of the operation of HTUs, the modeling method must be adaptable. Thus, the method should ensure the construction of models considering the individual parameters of HTUs. This is because of the high degree of uniqueness of each HTU, which is associated with the peculiarities of manufacturing, installation, operating conditions, duration, and the individual wear of the elements. This indicates the need to build a model for estimating parameters for each HTU, and the use of nonparametric approaches would be preferable for such conditions. Thus, the nonparametric approach allows for moving away from rigidly structured models, the creation of which is difficult taking into account the factors described above. In addition, the nonparametric approach considered here allows for integrating new data, obtained and verified by means of measurement and additional control, into the calculation model. Such reliable observations are simply included in the sample, which produces the calculated nonparametric model and ensures the adaptation of the model, taking into account newly obtained asynchronous measurements. In the course of the study, nonparametric Nadaraya–Watson kernel regression was used as the basis for the nonparametric model [14]. This method of nonparametric modeling is used quite successfully to build models and forecasting in many industries, including economics [15,16], medicine [17], energy [18,19], and the geosciences [20]. A brief description of the nonparametric kernel regression method is provided in Section 2.

The main issue in the application of the method of nonparametric modeling based on kernel regression estimates is the reasonable choice of the smoothing factor for calculating each of the bell-shaped functions of the nonparametric kernel. The corresponding problem can be considered as a multidimensional optimization problem, on the effective solution of which the quality of modeling depends significantly. A possible approach for solving this problem is the use of an efficient numerical optimization algorithm. For this purpose, the use of an evolutionary genetic algorithm with local search is proposed. The genetic algorithm is one of the most effective methods of global optimization, which found applications in many areas [21–24]. Complemented by local-search methods, the genetic algorithm is able to provide excellent properties for solving an optimization problem, similar to that formulated with respect to smoothing parameters for calculating a nonparametric regression model. A description of the proposed genetic algorithm with local search is also given in Section 2.

The use of the optimization algorithm makes it possible to form an additional adaptability contour, providing the versatility of the application of the approach under consider-

ation in conditions of the different density of measurements of the parameters of the HTU. Such adaptability, taking into account different ranges of variation of input parameters and the sampling rate of measuring devices, made it possible to improve the accuracy of modeling and the effectiveness of the approach in general. The results of a numerical study of the proposed nonparametric approach with optimization of the parameters of smoothing of bell-shaped functions by a genetic algorithm with local search are given in Section 3.

In the first part of the experimental study, the effectiveness of the proposed approach was evaluated on a set of test problems of regression modeling. To assess the added efficiency, studies were also carried out using a "pure" nonparametric model without the optimization of smoothing factors, as well as alternative approaches: artificial neural networks, as one of the most powerful tools that found application in various tasks of industrial signal processing [25,26]; genetic programming; and the method of multivariate regression splines. In the second part of the research, the accuracy of modeling and the stability of the obtained results were directly assessed on the data of the HTU's full-scale tests. A discussion of the results is given in Section 4.

## 2. Materials and Methods

### 2.1. Nonparametric Regression Estimation

The general formulation of the regression problem is as follows. Let there be observations of the input output variables of the process under study $V = (x_i, y_i); i = \overline{1, n}$, $n$ is the sample size of observations. There is a dependence $y = f(x)$ between the input and output variables of the process under study, but the type and structure of the dependence are not known.

It is necessary for available sample of observations $V$ to find a mathematical expression $\hat{y} = \hat{f}(x)$ that approximates the relationship between input and output variables.

As an optimality criterion, for example, the value of the relative average modeling error for the sample as a percentage can be used:

$$W = \frac{100\%}{n(y^{\max} - y^{\min})} \sum_{i=1}^{n} |\hat{y}(x_i) - y_i|, \tag{1}$$

where $i$ is the numerator; $i = \overline{1, n}$; and $y^{\max}$ and $y^{\min}$ are the maximal and minimal values of the output parameter, respectively.

To obtain a nonparametric regression estimate, it is necessary to estimate the unknown conditional distribution density:

$$p(y/x) = \frac{p(x, y)}{p(x)}, \tag{2}$$

where $p(x, y)$ is the joint probability density function of $x$ and $y$, and $p(x)$ is the probability density function of $x$. For this, the joint distribution density is estimated using the Rosenblatt–Parsen estimate [27]. The nonparametric regression estimate is calculated as follows:

$$\hat{y}(x) = \frac{\sum\limits_{i=1}^{n} y_i \Phi\left(\frac{x - x_i}{h(n)}\right)}{\sum\limits_{i=1}^{n} \Phi\left(\frac{x - x_i}{h(n)}\right)}, \tag{3}$$

where $\Phi$ is a truncated bell-shaped function (kernel), $(x_i, y_i)$ is $i$th point of the sample used to build the model, $h(n)$ is a smoothing factor, and $n$ is the sample size. The kernel regression estimate (3) is named the Nadaraya–Watson estimator. Theorems on the asymptotic properties of this estimate are proved. The main idea behind (3) is to give relatively more weight to the observations closest to the estimated point in the sense of the distance determined by the kernel.

Thus, it is possible to restore the relationship between the input and output of the object using the training sample of object observations to construct a nonparametric estimate. For the case of constructing a regression dependence on a set of input of parameters, Formula (3) becomes Formula (4):

$$\hat{y}\left(x^1,\ldots,x^m\right) = \frac{\sum\limits_{i=1}^{n} y_i \prod\limits_{j=1}^{m} \Phi\left(\frac{x^j - x_i^j}{h^j(n)}\right)}{\sum\limits_{i=1}^{n} \prod\limits_{j=1}^{m} \Phi\left(\frac{x^j - x_i^j}{h^j(n)}\right)}, \tag{4}$$

where, in addition to Formula (3), $m$ is the dimension of the vector of the input variables $x$.

A number of conditions are imposed on the kernel bell-shaped function and the smoothing factor, which ensure the convergence of the nonparametric estimate of the Nadaraya–Watson regression [27]. There are several types of bell-shaped functions that satisfy such conditions, such as rectangular, triangular, Epanechnikov, and Gaussian kernels. A commonly used kernel function is the Epanechnikov kernel [28]:

$$\Phi(z) = \begin{cases} 0.335 - 0.067z^2, & \text{if } z^2 \leq 5 \\ 0, & \text{if } z^2 > 5 \end{cases} \tag{5}$$

The accuracy of the regression model in practical conditions can be improved by selecting the optimal values of smoothing parameter $h(n)$. In the multidimensional case, the efficiency of the model is determined by the vector of parameters $h(n)$, each component of which determines the coverage area of the truncated kernel for the corresponding input variable—the components of vector $\vec{x}$.

The optimal value of the components of smoothing parameter vector $h(n)$ is found from the following relation:

$$h(n) = c \cdot n^{-1/5} \tag{6}$$

where $c$ is a positive constant. It is the choice of constant $c$ that determines the scaling of the smoothing factor relative to the calculated factor, which is determined on the basis of the sample size of observations, which has the greatest influence on the quality function.

Thus, in accordance with the dimension of the input feature space used to build the regression model, the vector of scaling constants of the bell-shaped functions must be determined to calculate smoothing factor $h(n)$. The performed numerical studies show that the rational choice of the corresponding constants for the vector of smoothing factors must be carried out for each task by minimizing the quality indicators (modeling errors) that characterize the best fit with the experimental data. In this regard, an additional optimization problem arises, on the solution of which the efficiency of solving the modeling problem as a whole significantly depends. The manual selection of smoothing parameters is laborious in the case of a high dimension of the feature space, thereby requiring the use of formal techniques based on effective optimization algorithms. A genetic algorithm is proposed to be used as such an algorithm, the application of which is described in the next section.

*2.2. Smoothing Factor: Optimization*

The approach described above to the construction of regression models can generally be characterized as nonparametric. The resulting model does not have a rigid fixed structure based on an analytical description of the modeled process or dependences of parameters. Accordingly, such a regression model does not contain a set of numerical parameters that determine, in conventional analytical models, the fit of the structure to the conditions of a specific application. The adjustable parameter in the regression model of form (3) is smoothing parameter $h(n)$. This parameter defines the scope of the kernel bell-shaped function, and should vary somewhat depending on the density and size of the sample used to build the regression model. Previous studies showed that the accuracy

of the built model largely depends on the value of smoothing parameter $h(n)$ [29]. The model is especially sensitive to this when the level of interference in the measurement channels increases, leading to noise in the original sample. As mentioned above, the optimal value of the smoothing factor in the sense of minimizing the quality indicator is determined by Formula (6). In the multidimensional case, the smoothing parameter must be defined for each component of the vector of input variables $x$. An essential difficulty is the choice of the optimal values of constants $c$. In the general case, taking into account the independence of the input parameters, as well as their significant difference in the measurement ranges and noisiness, it is necessary to ensure the selection of the corresponding constants independently of each other, and not to use the same value for all input parameters.

Thus, the optimization problem of multivariate optimization is structured, the target criterion of which is the function of evaluating the mismatch between the obtained regression model and the approximated sample data. Such a function must be minimized considering the selection of the optimal values of the smoothing parameters for each variable. Taking into account the impossibility of providing an analytical calculation of the solution to such a minimization problem, it can be solved by the method of numerical optimization. A possible variant of the method for solving the optimization problem can be the multiple use of one-dimensional optimization methods to separately determine the constants of the smoothing factors. Considering the experience of applying this approach, it is still rational to build such a procedure on the basis of the multidimensional optimization method, which ensures high efficiency in solving optimization problems for the functions of many variables.

### 2.3. Genetic Algorithm

On the basis of the experience of previous studies, the use of an evolutionary genetic algorithm as such a method was proposed. The genetic algorithm is a fairly powerful heuristic optimization tool that has gained wide acceptance and has a wide range of applications. The genetic algorithm as a method for optimizing the parameters of models and classifiers of various types has been repeatedly successfully tested in conditions of various applications. The breadth of the field of application, owing to the high efficiency of the approach, is presented in the works and reviews of many researchers of various orientations: engineering, medicine, biology, chemistry, and the material sciences [22–24,30].

Along with many other methods of global optimization, the genetic algorithm has both advantages and disadvantages. There are a huge number of different modifications that provide higher efficiency rates for solving particular optimization problems and improving the properties of the algorithm [31]. Within the framework of the present study, to improve the accuracy of the search for the values of the constants for calculating the vector of smoothing parameters of the regression model of form (3), it is proposed to combine the genetic algorithm with the local-search method. This combination makes it possible to increase the local convergence of the genetic algorithm at the final stage of the search for optimal solutions. Taking into account the rather high sensitivity of the quality criterion of the nonparametric regression model to the choice of the smoothing factor constants, the local-search method improves the efficiency of the optimization procedure as a whole. The following is a very short description of the genetic algorithm as an extended description can be found in various works devoted to optimization algorithms [21–24,30,31].

The standard scheme of the genetic algorithm assumes the coding of the variables of the optimization problem into a bit string that combines the components of the search space vector for any of the points, represented in binary. The set of points that are the current set of solutions to the problem is combined into a set called a generation. The initial generation is randomly created, and there are procedures for the step-by-step transformation of current generations into new generations. For this, the selection of potentially promising solutions (selection), the formation of new solutions on their basis (crossing), and a random change of new solutions with a given probability (mutation) are used. After performing

these operations, the current generation of "parents" is replaced by a new generation of "descendants" and the cycle is repeated. This happens until a satisfactory solution is found or a predetermined number of generations is produced. Despite its relative simplicity, even the standard genetic algorithm is a fairly effective global-search method. On the basis of such a basic structure of the genetic algorithm, various modifications are built aimed at expanding the class of problems to be solved, for example, multicriteria word-optimization problems, or at increasing its efficiency and improving some properties of the basic scheme.

*2.4. Genetic Algorithm with Local Search*

Within the framework of this work, to improve local convergence when adjusting the smoothing factors for the regression model, it is proposed to modify the standard genetic algorithm scheme by adding a local step-by-step random search stage. The practice of using the genetic algorithm as an optimization algorithm with adjustable parameters has shown that the greatest difficulty in setting parameters is a contradiction: global search versus local convergence. Researchers have several tools at their disposal to shift the balance in one direction or another, but it is practically impossible to track the degree of this balance shift. This leads to the fact that it is often very difficult and sometimes almost impossible to tune the standard genetic algorithm in such a way that some "golden mean" in the considered contradiction was found.

Achieving greater efficiency in solving problems in this case is possible owing to the use of a hybrid genetic algorithm. A hybrid algorithm is a standard genetic algorithm supplemented by one of the local-search methods [32]. At the same time, local search is carried out both after the completion of the genetic algorithm and at each generation to improve a certain number of individuals. Usually, local search is used for the most promising individuals of the generation.

Just as the standard genetic algorithm has parallels with the processes occurring in living nature, so does the hybrid algorithm model the lifetime adaptation of individuals in the population. In binary space, the search step is carried out as follows:

- The selected position of the binary string encoding the selected solution (the so-called gene) is inverted.
- The value of the fitness function of the modified solution, which was changed in the previous step, is calculated.
- If the suitability of the modified solution is higher than the suitability of the original solution, then the search step is considered to be successful, and the new value of the gene is fixed. Otherwise, the search step is considered to be unsuccessful, and the original value is returned to the gene.
- A new local-search step if the number of steps does not exceed the specified number; otherwise, the search stops and the modified solution returns to the original set (population).

The considered procedure of local search cannot lead to a deterioration in the fitness of an individual.

Thus, because of the use of a genetic algorithm hybridized by local search, it is proposed to implement a scheme for constructing nonparametric models with the optimization of smoothing parameters. An overview of such a scheme is shown in Figure 1.

The convergence of this method is ensured by the fundamental properties of the genetic algorithm as an optimization procedure for global search with proven convergence for functions of various types, including nondifferentiable ones. The convergence of the procedure for adjusting the coefficients using an additional local-search procedure is not violated as the local-search procedure cannot worsen the properties of solutions generated by the genetic algorithm. If the local-search procedure fails, the original individual produced by the genetic algorithm is returned to the generation.

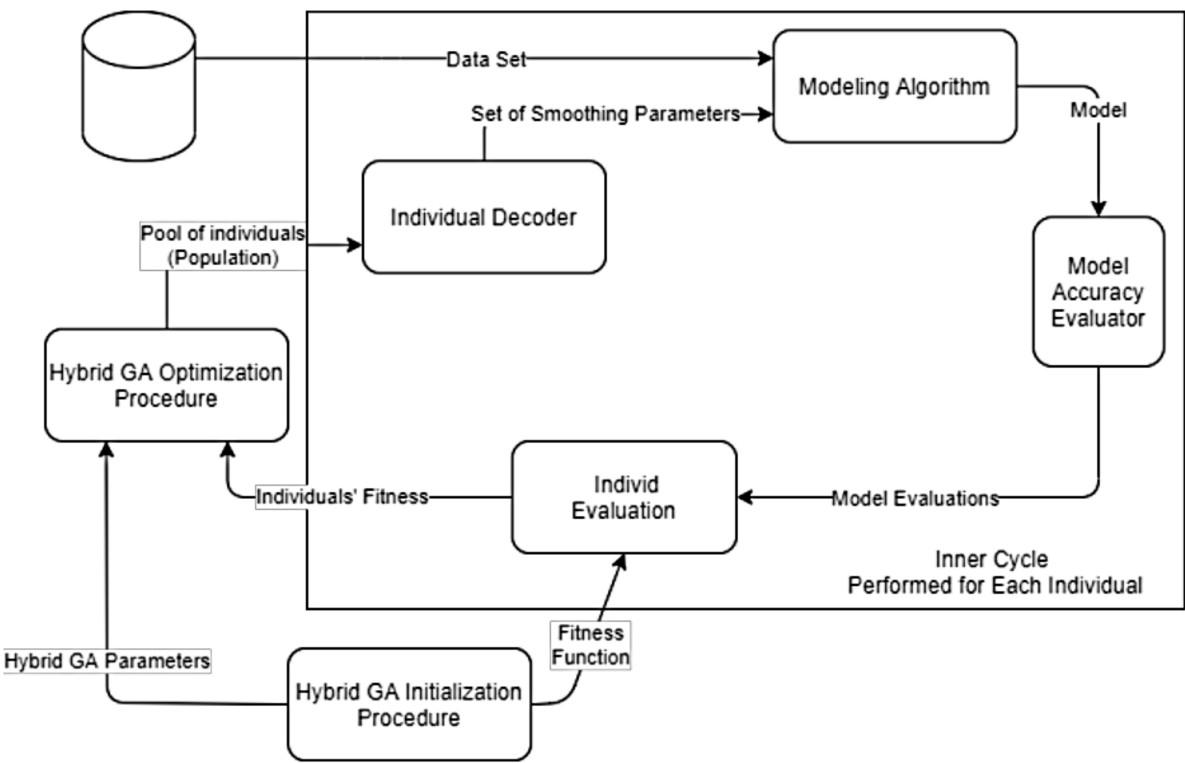

**Figure 1.** Smoothing-factor optimization scheme.

*2.5. Experimental Study*

2.5.1. List of Test Problems

The evaluation of the feasibility and comparative study of the effectiveness of the approach were carried out in the course of numerical studies. To conduct a preliminary study of the possibility of applying the proposed approach to the construction of non-parametric regression models with adjusting smoothing factors, we formed a set of test functions commonly used for evaluating the basic efficiency of modeling approaches. A brief description of the functions included in the test set is given in Table 1.

**Table 1.** Set of test functions.

| No. | Simulated Function | Input Variables |
|---|---|---|
| 1 | $y = \sin x$ | $x \in [-4, 3]$ |
| 2 | $y = x_1^2 \sin x_1 + x_2^2 \sin x_2$ | $x_i \in [-4, 3]$ |
| 3 | $y = \frac{x_1 \cdot x_2}{x_2^2}$ | $x_i \in [1, 20]$ |
| 4 | $y = 100(x_2 - x_1^2)^2 - (1 - x_1)^2$ | $x_i \in [-2, 3]$ |
| 5 | $y = \frac{\sin|x|}{|x|}$ | $x \in [-2\pi, 2\pi]$ |
| 6 | $y = \frac{\sin \sqrt{x_1^2 + x_2^2}}{\sqrt{x_1^2 + x_2^2}}$ | $x_i \in [-4\pi, 4\pi]$ |
| 7 | $y = 10\sin(\pi x_1 x_2) + 20(x_3 - 0.5)^2 + 10x^4 + 5x^5$ | $x_i \in [0, 1]$ |
| 8 | $y = \sqrt{x_1^2 + \left(x_2 x_2 - \frac{1}{x_2 x_4}\right)^2}$ | $x_1 \in [0, 100]$ $x_2 \in [40\pi, 560\pi]$ $x_3 \in [0, 1]$ $x_4 \in [1, 11]$ |
| 9 | $y = \frac{\pi}{2}\exp\left[-2\left(x_1^2 + x_2^2\right)\right]\cos[2\pi(x_1 + x_2)]$ | $x_i \in [0, 1]$ |
| 10 | $y = 0.79 + 1.27x_1 x_2 + 1.56x_1 x_2 + 3.42x_2 x_5 + 2.06x_3 x_4 x_5$ | $x_i \in [0, 1]$ |

Such a set of test problems is not exhaustive, but it was used by the authors in previous studies and showed the suitability for assessing the basic efficiency of approaches to constructing models of various types. In addition, for previously studied methods, a

fairly stable correlation was observed between the obtained results using this test set and the quality of modeling for problems with real data.

The study on the test set also allows for us to assess the stability of the proposed approach in the conditions of the formation of nonparametric models for samples of different dimensions and in conditions of overlapping noise of different levels. The variation of the corresponding parameters is given in Section 2.5.1.

### 2.5.2. HTU Data

Modern HTUs are complex technological units, the safety and efficiency of which are the main priorities in their operation. Such equipment is characterized by a number of properties that actualize the need to build effective computational models to assess parameters characterizing their operational efficiency and technical condition. The reliability of determining parameters of the technical condition is one of the basic needs in ensuring the safety of operation. The importance of forming an approach for constructing effective computational models is because each HTU is a unique product of mechanical engineering. In addition to this, each hydraulic unit has features of installation, positioning, and operation. That is, each stage of the HTU life cycle of even one model line or one HTU hall is individual, which obviously affects the patterns and relationships underlying the measured and target design parameters of functioning efficiency, and the parameters that determine the reliability and safety of operation. One of the main directions of such parametric analysis of the operation of HTUs is based on the use of a set of operational indicators and vibration parameters as input parameters of computational models. The vibration parameters determined at the fixed measurement points for such rotating equipment as a HTU are an essential informative source for determining the calculated indicators of efficiency and safety. Their inclusion in the integrated model, along with the measured parameters, makes it possible to fully assess the current state of the HTU, if necessary, to correct the operating modes, and rationally plan the operational and repair periods.

In order to briefly explain the source of obtaining the initial sample parameters, measured directly on the HTU, we present a diagram of an HTU with a typical arrangement of measuring sensors in Figure 2.

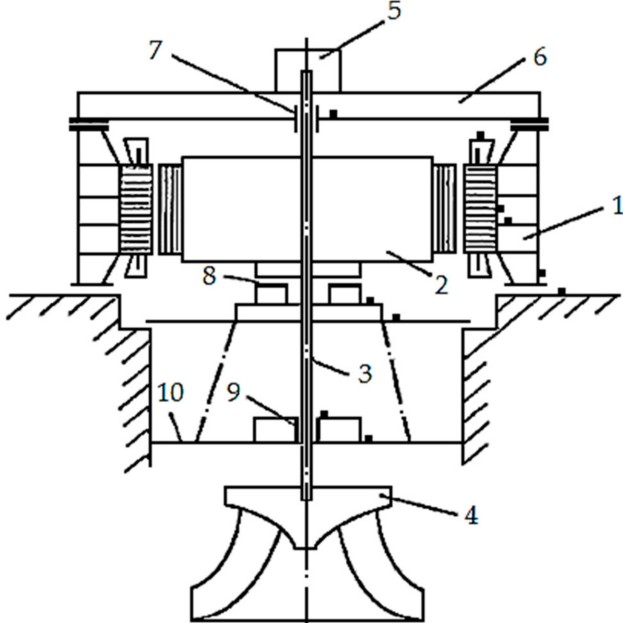

**Figure 2.** Hydroturbine unit (HTU) with typical arrangement of measuring sensors: 1, generator stator; 2, rotor; 3, hydraulic unit shaft; 4, hydraulic turbine; 5, exciter; 6, upper crosspiece; 7, upper guide bearing; 8, thrust bearing; 9, turbine bearing; 10, turbine cover; ■, installations of measuring transducers.

The total number of measuring transducers is determined by the characteristics of a particular HTU, which is monitored to collect diagnostic information. The generalized diagram contains a number of measuring transducers that are excessive for this study, some of which are intended to bring the monitoring system to the requirements of regulators. To carry out numerical studies in the monitoring process, the following 10 parameters were simultaneously measured: radial vibration of the turbine bearing housing (μm); radial vibration of the generator bearing housing (μm); shaft runout in the area of the turbine bearing (μm); shaft runout in flange connection (mm); vertical vibration of the turbine cover (μm); vertical vibration of the upper cross (μm); unit capacity (MW); upstream and downstream levels (m); guide vane opening (mm); and blade rotation angle (degrees). The following were considered to be output parameters: dynamic force on the turbine bearing, dynamic force on the generator bearing, dynamic force on the thrust bearing (turbine cover) in the axial direction, and the total load on the bearings (indicated as HTU Param 1–4, respectively, in Section 3).

Thus, the dataset used in the study includes 1280 observations, each of which is described by 10 input and 4 output parameters. The output parameters make it possible to comprehensively assess the state of the investigated object (HTU); however, in real-life operation, monitoring some of them is impossible, and some of the measurements are extremely expensive. Therefore, the creation of models describing the relationship between input and output parameters, allowing for simulating the output parameters for the 10 inputs monitored in real time, is an urgent task. Obviously, when using adequate models, fixing the moments when the simulated values of the output parameters exceed the critical values prevents the development of emergency situations and the occurrence of security incidents.

2.5.3. Numerical Experiment Technique

For a preliminary study of the nonparametric regression method on test functions from the set described above, samples of input variables were generated, and the values of the output parameters were calculated for the corresponding functions. For a comprehensive assessment of the adaptability of the approach, the samples were formed five times for each value according to the number of points in the samples and the level of imposed interference. Additive noise formed in accordance with Gaussian normal distribution with a mathematical expectation equal to 0 and a standard deviation equal to the specified percentage of the variation interval of the corresponding function was used as noise. Sample-generation parameters for test functions from the considered set are shown in Table 2.

**Table 2.** Parameters of samples for test functions.

| Sample parameter | Unit | Values |
| --- | --- | --- |
| Sample size | Points | 20, 50, 100, 250, 1000 |
| Additive noise level | Percentage of function-variation level | 0, 5, 10, 25 |

Because, in contrast to the test problems, the available sample for the HTU was fixed, to ensure the correctness of the results of the study of methods, the sample was repartitioned into a training sample used to build a model and a test sample used to obtain the final estimates of the adequacy of the corresponding models.

To ensure the unity of the numerical experimenting scheme, such repartitioning was performed five times in the following proportion: 85% of the sample was used directly to build and adjust the model, and 15% of the original sample was used as a test sample to calculate the values entered in the table of results. Taking into account the selected distributions during each repartitioning, the training sample was formed in the amount of 1090 patterns, and the test sample in the amount of 190 patterns. The error in each run was calculated using the quality criterion of the regression model determined by Formula (1).

For the settings of other methods considered in the course of numerical studies, the following parameters were determined. For the method of artificial neural networks, the automatic generation of networks with multilayer-perceptron architectures and network architecture with radial basis functions was used. The maximal number of neurons on hidden layers was set equal to 30, and the maximal number of hidden layers for a multilayer perceptron was chosen to be 2. The sample was split into training and test samples in accordance with the above scheme in the proportion of 85% and 15%, respectively, with multiple repartitioning and cross-validation. The same scheme for forming the tuning and test samples and validating the results was used for the method of symbolic regression (genetic programming). The construction of a symbolic regression model was carried out using the following parameters of the evolutionary process: the maximal number of generations was 200, the number of individuals in a generation was 50, the mutation was standard, and the crossing was one-point. The maximal depth of the symbolic regression tree was set to 12 levels. The method of multivariate adaptive regression splines was used in accordance with the standard implementation scheme in applied software package for statistical analysis Statistica.

The tables of results below show the estimates of the mean value obtained for each sample on the basis of the results of running the modeling algorithm of nonparametric regression five times. Taking into account the probabilistic initialization of solutions in the used genetic algorithm for adjusting the smoothing factors, and thereby obtaining stable results, a $5 \times 5$ numerical research scheme was used. That is, to obtain each value, 25 values were used, obtained in the course of separate numerical experiments (runs).

For all methods, in order to obtain correct results of numerical experiments, the same limitation on the amount of computational resources available to the method for obtaining a solution was used. This was limited at the level of assessing the use of CPU time owing to the use of a software timer. ANOVA methods were used to investigate the statistical significance of differences in the effectiveness of the used approaches. A pairwise comparison of the investigated methods was carried out to reveal statistical significance in the distinguishability of the obtained results during testing at the significance level of 0.05.

### 2.5.4. Genetic Algorithm Parameters

To use the genetic algorithm, it is necessary to select the parameters that control the process of finding solutions. This task is quite capacious for research and goes beyond the scope of this study. For use here, the parameters of the genetic algorithm were selected in the course of a preliminary study on a representative set of functions. The main parameters of the genetic algorithm include the number of generations and the size of the population, the type of selection, the type of crossing, the probability of mutation, and the proportion of substitution in a generation.

The number of generations and the size of the population were selected to provide similar computation times on average compared with the alternative modeling approaches discussed in this article. The population size was limited to 50 individuals, and the maximal number of generations of tuning parameters was 200 or until the stop criterion for no improvement in the solution was met for 25 generations.

The type of selection was defined as tournament selection with a tournament size of 5, the type of crossing was single-point, the type of mutation was medium, and the percentage of replacement in a generation was 90%. Such a set of parameters was chosen on the basis of the average highest efficiency of the algorithm when running along test functions.

### 3. Results

Tables 3–5 show the results of a comparative study of the proposed nonparametric regression method with the adaptation of smoothing factors using a genetic algorithm and a genetic algorithm with local search to solve this problem. Criterion (1) was used for comparison.

**Table 3.** Results of the study of nonparametric models with a random value of smoothing factors on the test set.

| Noise Level | Sample Size | | | | |
|---|---|---|---|---|---|
| | 20 | 50 | 100 | 250 | 1000 |
| | Mean Modelling Error, % | | | | |
| 0 | 9.8 | 9.6 | 8.8 | 7.8 | 7.2 |
| 5 | 10.2 | 10.0 | 9.2 | 8.1 | 7.8 |
| 10 | 10.6 | 10.4 | 10.1 | 8.4 | 8.2 |
| 25 | 16.3 | 14.8 | 13.9 | 11.1 | 10.6 |

**Table 4.** Results of the study of nonparametric models by adjusting smoothing factors with a standard genetic algorithm (GA) on a test set.

| Noise Level | Sample Size | | | | |
|---|---|---|---|---|---|
| | 20 | 50 | 100 | 250 | 1000 |
| | Mean Modelling Error, % | | | | |
| 0 | 5.2 | 3.5 | 2.9 | 1.8 | 1.0 |
| 5 | 5.4 | 3.6 | 3.0 | 1.9 | 1.5 |
| 10 | 5.6 | 3.7 | 3.1 | 2.0 | 1.8 |
| 25 | 6.1 | 4.1 | 3.7 | 2.6 | 2.5 |

**Table 5.** Results of the study of nonparametric models by adjusting smoothing factors by a hybrid GA with local search on a test set.

| Noise Level | Sample Size | | | | |
|---|---|---|---|---|---|
| | 20 | 50 | 100 | 250 | 1000 |
| | Mean Modelling Error, % | | | | |
| 0 | 3.9 | 2.7 | 1.5 | 1.3 | 0.7 |
| 5 | 4.1 | 2.8 | 1.6 | 1.4 | 0.8 |
| 10 | 4.2 | 2.9 | 1.7 | 1.4 | 0.8 |
| 25 | 4.7 | 3.6 | 2.7 | 2.1 | 1.5 |

The method of random selection of constants $c$ for calculating the smoothing factor in accordance with Formula (6) was implemented as a basic option for assessing the effectiveness of the methods for adjusting the smoothing factors. The random generation of values was carried out in the same range of [0.01; 100], which was also used when searching for solutions using the methods of the genetic algorithm and the genetic algorithm with local search. The results for the baseline approach are shown in Table 3.

The next stage of numerical research was the evaluation of the method that used a standard genetic algorithm to find the optimal set of constant values for calculating the smoothing factors for each of the input variables. The search range was chosen to be [0.01; 100] for each parameter, the number of which corresponds to the dimension of the input space of the test function. The accuracy of the presentation of the solutions (maximal sampling step) was determined to be 0.01. Thus, the number of sampling steps of the search along one dimension of space was 104. In accordance with the decision-coding scheme in the genetic algorithm in the form of a bit string, a sequence was determined to encode one constant of the smoothing factor in the size of 14 bits. The set of constants that determined the smoothing factors for all the variables of the problem was coded as a concatenation of the bit sequences of each constant, decoding the solution in a standard way for the genetic algorithm. The results of a statistical study of the quality of constructing nonparametric models with the adjustment of the smoothing factors by the genetic algorithm are shown in Table 4.

Further, studies were carried out suggesting the use of an additional procedure that provides a higher local convergence of the genetic algorithm. In general, the main stages are performed in full accordance with the scheme of the standard genetic algorithm. In addition to this, at every 10th step (preliminary estimate for test optimization problems), a local-search procedure was performed to refine solutions and search for improved options in the vicinity of points already formed by the basic procedure. The preservation of parity in the use of computing resources was ensured by a proportional reduction in the population size by 10% to 45 individuals in comparison with the basic genetic algorithm, the parameters of which are given above. The results of applying such a hybrid scheme of the genetic algorithm and local search to adjust the smoothing factors of nonparametric models are shown in Table 5.

Studies carried out on a set of test functions showed the possibility of a significant increase in the adequacy of the regression model by adjusting the smoothing factors. At the same time, the use of local search in addition to the global search method—the genetic algorithm—made it possible to find an even more efficient set of smoothing factors for bell-shaped functions. For a more detailed discussion of the results, see Section 4.

The next stage, after confirming the operability and increasing the efficiency of the nonparametric regression approach, was its comparative study with other methods of constructing regression models on the problem of estimating the parameters of a HTU by vibration characteristics considered in the study. In accordance with the scheme of numerical studies described above, estimates of errors were obtained, calculated with Formula (1) for samples, one of which was directly used for constructing the model, and the second was used to assess the quality of the constructed model. Regression models were built for each of the four output parameters; their construction was carried out in accordance with the accepted scheme of numerical studies. The results were averaged over the results of multiple model building. The corresponding values of the estimates of the modeling error are shown in Table 6.

**Table 6.** Results of the study of modeling approaches on problem of estimating HTU parameters.

| Modeling Approach | Mean Modelling Error, % | | | |
|---|---|---|---|---|
| | HTU Param 1 | HTU Param 2 | HTU Param 3 | HTU Param 4 |
| Nonparametric regression (hybrid-genetic-algorithm tuning) | 6.9 | 5.0 | 7.5 | 6.1 |
| Artificial neural networks | 13.8 | 13.6 | 11.1 | 14.2 |
| Regression by genetic programming | 17.2 | 16.8 | 18.5 | 18.4 |
| Multivariate adaptive regression splines | 8.8 | 6.1 | 8.8 | 7.6 |

The results for each of the series of numerical experiments were tested for statistical significance, which was confirmed in the course of testing the corresponding hypotheses.

## 4. Discussion

The results of the numerical study of efficiency on a set of test functions confirmed the possibility and necessity of formalizing the approach to the choice of smoothing coefficients in the case of using the nonparametric regression method. An arbitrary choice of a constant that determines the value of the smoothing factor in the case of failure can lead to a significant decrease in the effectiveness of the proposed approach, which is accompanied by an increase in modeling error. Consequently, it is necessary to rationally approach the choice of the appropriate value for the smoothing factor, but its manual selection, especially in the case when the dimension of the space of input variables increases, is difficult. Statistical studies showed that, even for functions used with the number of input variables from 1 to 4, the use of smoothing factor adjustment algorithms makes it possible to achieve a significant refinement of the nonparametric regression model. The implementation of such a procedure using the global-search method, a genetic algorithm, provided a

reduction in modeling error by about 20% compared with the method of multiple random selection of the smoothing factors. Taking into account the high sensitivity of the model quality criterion to the choice of the smoothing factor, an approach that provides the local refinement of smoothing factors constants is even more effective. This was implemented using a hybrid genetic algorithm, which improved the model's accuracy by up to 10% compared with that of the standard genetic algorithm, and up to 30% compared with multiple random enumeration.

In terms of the results of a numerical study of the effectiveness of the proposed approach for assessing the parameters of an HTU, the higher accuracy of nonparametric models with the adjustment of blurring factors was statistically confirmed. The most effective for solving this problem were the computational models—the nonparametric model and the model of multidimensional adaptive regression splines. Between them, the differences in the results were about 10%–15% in terms of the magnitude of the modeling error, which is a fairly good result. However, the software used showed less computational load when constructing the multivariate adaptive spline model. Thus, a standard trade-off was demonstrated between the accuracy of the model and the time of its construction, the priorities at the resolution of which are selected from the conditions of a particular application. The advantage of the nonparametric regression approach is stability in processing noisy data. The results of neural-network modeling were somewhat disappointing, but this may have been because of the peculiarity of using networks with a fixed structure of multilayer perceptrons and networks with radial-basis functions. In further studies, it is proposed to use neural networks with an automated procedure for forming a structure as a more powerful alternative [33]. Taking into account the experience of the application, it is proposed to form ensembles of models including neural networks for their ability to generalize and precise nonparametric procedures that ensure the local efficiency of the models.

The symbolic regression method based on genetic programming made it possible to form less efficient models on average in terms of approximation accuracy. Nevertheless, the models built in this way can be considered as basic analytical parametric dependencies, the generation of which in an automated mode is a complex and demanded problem. It seems that it is precisely the combined analytical cores based on models of various classes that can be one of the solutions for the creation of integrated systems for the computational processing of production data for HTU. They make it possible to, on the one hand, solve the problem of the fast and accurate calculation of the values of operational parameters and, on the other hand, form support for decisions in the search for cause-and-effect relationships when analyzing the features of the operation of a specific HTU. This is provided by a combination of the models evaluated in this study, which demonstrated the features of their application in solving such a problem.

## 5. Conclusions

In the course of this study and its results, the problem of constructing models for calculating the parameters of the functioning of a HTU was solved. The operating conditions of such equipment, and restrictions imposed on the means of measuring and monitoring the parameters of the technical condition of the HTU, have actualized the problem of constructing computational models for use in the integrated monitoring system and ensuring operational safety. Taking into account the high variability of the parameters of HTUs, owing to the individual characteristics of the passage of such equipment through the stages of the life cycle, nonparametric regression models were used as basic models in the study. It was estimated that the quality of such models is largely determined by the effectiveness of the selection of smoothing factors that determine the coverage area of the basic bell-shaped functions. In this regard, a procedure was proposed for finding the optimal values of such parameters on the basis of an evolutionary genetic algorithm. To improve the accuracy of determining the parameters, in addition to the standard optimization scheme using the genetic algorithm, the local-search stage was also proposed for use. The corresponding

procedure can be characterized as a hybrid genetic algorithm. The study of the basic genetic algorithm and the proposed hybrid scheme for choosing the smoothing factors when restoring test functions from a test set was carried out. The high efficiency of the improved hybrid schemes for optimizing smoothing factors for nonparametric modeling was shown. Taking into account the insignificant increase in computational complexity that was leveled out by reducing the population in relation to the basic scheme of the genetic algorithm, it is reasonable to further use the hybrid genetic algorithm for such problems.

Then, the approach was directly used for the formation of models for calculating parameters characterizing the technical condition of the HTU. As alternatives to the main approach under consideration, the parametric method was also investigated—the method of reconstructing symbolic regression by genetic programming, modeling based on artificial neural networks, and the method of multivariate adaptive regression splines. Analysis of the obtained results confirmed the relatively high efficiency of the nonparametric method with the adjustment of the smoothing factors by the hybrid genetic algorithm. On average, relative to other approaches, the nonparametric regression model provides 10%–15% reduction in modeling error. Such a result, taking into account the confirmation of its statistical significance, can be considered to be essential for the problem under consideration.

In the future, it is possible to build combined models that provide the possibility of both fast nonparametric calculation and the determination of functional dependencies. This would make it possible to ensure the greater stability of the results of model calculations within the framework of an integrated decision-support system for the operation of HTUs.

**Author Contributions:** Conceptualization, V.V.B.; Data curation, V.V.B. and V.S.T.; Formal analysis, V.S.T. and V.V.B.; Investigation, V.S.T. and V.V.B.; Methodology, V.V.B.; Project administration, V.V.B.; Resources, V.V.B. and V.S.T.; Software, V.V.B. and V.S.T.; Supervision, V.S.T.; Validation, V.S.T.; Visualization, V.S.T.; Writing—original draft V.V.B.; Writing—review and editing, V.V.B. and V.S.T. All authors have read and agreed to the published version of the manuscript.

**Funding:** The reported study was partially funded by the Scholarship of the President of the Russian Federation for young scientists and graduate students SP.869.2019.5.

**Institutional Review Board Statement:** Not applicable.

**Informed Consent Statement:** Not applicable.

**Data Availability Statement:** HTU data set available at https://yadi.sk/i/NvZOMjXleCg71Q, accessed on 28 June 2021.

**Conflicts of Interest:** The authors declare no conflict of interest.

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
