# Peer review of "Design of Computational Models for Hydroturbine Units Based on a Nonparametric Regression Approach with Adaptation by Evolutionary Algorithms"

_computation, doi:10.3390/computation9080083_

Round 1
Reviewer 1 Report
This article represents a computational design to establish a non-parametric regression modeling for estimating the operational input (10 parameters) and output (4 parameters) for a specific Hydro-Turbine Unit, by using an advanced algorithm with adjustable smoothing factor. The works in this paper possess a high practical value for relevant engineering equipment.
Recommendation: Minor revision.
Specific comments:
- Since this study refers to an application of proposed advanced algorithm into engineering measurement data for Hydro-Turbine Unit, some figures and plots may supportthe readers to have a better understanding about the measurement and workflow chart of this paper.
- Abstract, “The carried out statistical studies have shown that the considered methods in general make it possible to obtain 5-15% more accurate values of the estimated parameters in comparison with the basic computational model ”. The basic computational model is not clearly mentioned in this study.
- Abstract, “The reduction in modeling error for the proposed nonparametric approach with a hybrid smoothing coefficient tuning scheme was found to be most effective with a reduction in modeling error by about 7% compared to the best alternative approach”. Again it is not clear about best alternative approach.
- The key wordsin this paper are not appropriate to summarize the core content and significance of this article in brief.
- Introduction, the author generally described the paper structure (section 2-5) from 2nd to 6th paragraph, but the section number mentioned in those paragraphs is unmatched with the following structure throughout this paper. Please check and correct it.
- Section for denote of Equations (2), “ymax and ymin are the minimum and maximum values of the output parameter, respectively”, please correct the corresponding values.
- Section 2. for denote of Equations (4), “where in addition to formula 3 m is the dimension of the vector of the input variables x”, it missed a comma symbolbetween 3 and m.
- Section 3. The criterion (**) and formula (*) mentioned in 1stand 2nd paragraphs are unclear. It is better to explain them in detail.
- There is no sufficient description (training and test procedure, sample size, etc) about other modeling approaches (such as artificial neural networks) involved in Table 6.
- There are many other typosin the paper: such as 1), there is no formula (7) in this paper, but it appears in last paragraph of section 3; ii), Conclusion section should be Section 5, not 4.
Author Response
Dear Reviewer 1,
Thank you for your positive comments on our paper. You provided remarks about it, so we’ve made changes in the paper.
Remark 1: Since this study refers to an application of proposed advanced algorithm into engineering measurement data for Hydro-Turbine Unit, some figures and plots may supportthe readers to have a better understanding about the measurement and workflow chart of this paper
Reply: Thank you for the comment. We introduced figures 1 and 2 in relevant sections for this purpose in revised version.
Remark 2: Abstract, “The carried out statistical studies have shown that the considered methods in general make it possible to obtain 5-15% more accurate values of the estimated parameters in comparison with the basic computational model ”. The basic computational model is not clearly mentioned in this study.
Reply: Thank you for the comment. The statement was deleted from the abstract due to the absence of the basic computational model in the version submitted for publication.
Remark 3: Abstract, “The reduction in modeling error for the proposed nonparametric approach with a hybrid smoothing coefficient tuning scheme was found to be most effective with a reduction in modeling error by about 7% compared to the best alternative approach” Again it is not clear about best alternative approach.
Reply: Thank you for the comment. We clarified that we were talking about the method of multivariate adaptive regression splines in the revised version of the paper.
Remark 4: The key wordsin this paper are not appropriate to summarize the core content and significance of this article in brief. 5.
Reply: Thank you for the comment. We tried to make it better in the revised version of the paper.
Remark 5: Introduction, the author generally described the paper structure (section 2-5) from 2nd to 6th paragraph, but the section number mentioned in those paragraphs is unmatched with the following structure throughout this paper. Please check and correct it
Reply: Thank you for the comment. We corrected it in the revised version of the paper.
Remark 6: Section for denote of Equations (2), ymax and ymin are the minimum and maximum values of the output parameter, respectively”, please correct corresponding values
Reply: Thank you for the comment. We corrected it in the revised version of the paper.
Remark 7: Section 2. for denote of Equations (4), “where in addition to formula 3 m is the dimension of the vector of the input variables x”, it missed a comma symbol between 3 and m.
Reply: Thank you for the comment. We added comma symbol between 3 and m.
Remark 8: Section 3. The criterion (**) and formula (*) mentioned in 1 and 2 paragraphs are unclear. It is better to explain them in detail.
Reply: Thank you for the comment. We replaced * and ** signs with appropriate numbers
Remark 9: There is no sufficient description (training and test procedure, sample size, etc) about other modeling approaches (such as artificial neural networks) involved in Table 6.
Reply: Thank you for the comment. We introduced a brief description about other modelling approaches:
As for the settings of other methods considered in the course of numerical studies, the following parameters were determined for them. For the method of artificial neural networks, automatic generation of networks with multilayer perceptron architectures and network architecture with radial basis functions was used. The maximum number of neurons on hidden layers was set equal to 30, the maximum number of hidden lay-ers for a multilayer pereceptron was chosen to be 2. The sample was split into training and test samples in accordance with the above scheme in the proportion of 85 and 15 percent, respectively, with multiple repartitioning and cross-validation. The same scheme for forming the tuning and test samples and validating the results was used for the method of symbolic regression (genetic programming). The construction of a symbolic regression model was carried out using the following parameters of the evolutionary process: the maximum number of generations is 200, the number of individuals in a generation is 50, the mutation was standard, and the crossing was one-point. The maximum depth of the symbolic regression tree was set to 12 levels. The method of multivariate adaptive regression splines was used in accordance with the standard implementation scheme in the applied software package for statistical analysis Statis-tica.
Remark 10: There are many other typosin the paper: such as 1), there is no formula (7) in this paper, but it appears in last paragraph of section 3; ii), Conclusion section should be Section 5, not 4.
Reply: Thank you for the comment. We corrected it in the revised version of the paper.
With best regards,
Dr. Vadim Tynchenko

Reviewer 2 Report
The paper is very good and deserves publication in Energies. The topic is a cogent one and the paper represents an important contribution.
The title is descriptive. The abstract clearly indicates the scope. The paper is well organised and logically written, nevertheless, the English language of the contribution should be improved by a native speaker.
Appropriate research goals are chosen in this contribution, which shows that the authors have a good level of understanding of current research within the field. The authors have been able to draw logical conclusions from the results.
The presentation of the results in terms of the research objectives has been made, nevertheless one clarification should be done before publishing.
- Please discuss more in Depth the tuning "philosophy" with respect to the non convexity of the considered Problem and how the Tuning strategy improves the convergence of the proposed method.
- Concerning the literature also the following papers can improve the General Background of this cvery goob contribution.
-
Mercorelli, P. Denoising and harmonic detection using nonorthogonal wavelet packets in industrial applications (2007) Journal of Systems Science and Complexity, 20 (3), pp. 325-343. Nentwig, M. et al. Throttle valve control using an inverse local linear model tree based on a Fuzzy neural network (2008) 2008 7th IEEE International Conference on Cybernetic Intelligent Systems, CIS 2008,
Author Response
Dear Reviewer 2,
Thank you for your positive comments on our paper. You provided remarks about it, so we’ve made changes in the paper.
Remark 1: The title is descriptive. The abstract clearly indicates the scope. The paper is well organised and logically written, nevertheless, the English language of the contribution should be improved by a native speaker. Appropriate research goals are chosen in this contribution, which shows that the authors have a good level of understanding of current research within the field. The authors have been able to draw logical conclusions from the results.
Reply: Thank you for the positive comments on the paper. As for English language in the paper, we will get an English editing service via MDPI.
Remark 2: The presentation of the results in terms of the research objectives has been made, nevertheless one clarification should be done before publishing. Please discuss more in Depth the tuning "philosophy" with respect to the non convexity of the considered Problem and how the Tuning strategy improves the convergence of the proposed method
Reply: Thank you for the comment. We added graphics and text in the end of subsection 2.4 trying to prove descript and prove convergence of the approach briefly.
The considered procedure of local search cannot lead to a deterioration in the fitness of an individual.
Thus, due to the use of a genetic algorithm hybridized by local search, it is proposed to implement a scheme for constructing nonparametric models with optimization of smoothing paraeters. An overview of such a scheme is shown in Figure 1 "Smoothing Factor Optimization Scheme".
The convergence of this method is ensured by the fundamental properties of the genetic algorithm as an optimization procedure for global search with proven convergence for functions of various types, including non-differentiable ones. The convergence of the procedure for adjusting the coefficients using an additional local search procedure is not violated since the local search procedure cannot worsen the properties of solutions generated by the genetic algorithm. If the local search procedure fails, the original individual produced by the genetic algorithm is returned to the generation.
Remark 3: Concerning the literature also the following papers can improve the General Background of this very good contribution.
Reply: Thank you for the comment. We have read the papers and included them into reference list with appropriate citation in the text.
With best regards,
Dr. Vadim Tynchenko
